# Cyanobacterial Abundance and Microcystin Profiles in Two Southern British Lakes: The Importance of Abiotic and Biotic Interactions

**DOI:** 10.3390/toxins12080503

**Published:** 2020-08-05

**Authors:** David M. Hartnell, Ian J. Chapman, Nick G. H. Taylor, Genoveva F. Esteban, Andrew D. Turner, Daniel J. Franklin

**Affiliations:** 1The Centre for Environment, Fisheries and Aquaculture Science (Cefas), The Nothe, Barrack Road, Weymouth, Dorset DT4 8UB, UK; nick.taylor@cefas.co.uk (N.G.H.T.); andrew.turner@cefas.co.uk (A.D.T.); 2Centre for Ecology, Environment and Sustainability, Faculty of Science & Technology, Bournemouth University, Fern Barrow, Poole, Dorset BH12 5BB, UK; ijchapman@outlook.com (I.J.C.); gesteban@bournemouth.ac.uk (G.F.E.); dfranklin@bournemouth.ac.uk (D.J.F.); 3New South Wales Shellfish Program, NSW Food Authority, Taree 2430, Australia

**Keywords:** flow cytometry, liquid chromatography coupled to tandem mass spectrometry (LC-MS/MS), cyanotoxins, risk assessment, management strategies, modelling

## Abstract

Freshwater cyanobacteria blooms represent a risk to ecological and human health through induction of anoxia and release of potent toxins; both conditions require water management to mitigate risks. Many cyanobacteria taxa may produce microcystins, a group of toxic cyclic heptapeptides. Understanding the relationships between the abiotic drivers of microcystins and their occurrence would assist in the implementation of targeted, cost-effective solutions to maintain safe drinking and recreational waters. Cyanobacteria and microcystins were measured by flow cytometry and liquid chromatography coupled to tandem mass spectrometry in two interconnected reservoirs varying in age and management regimes, in southern Britain over a 12-month period. Microcystins were detected in both reservoirs, with significantly higher concentrations in the southern lake (maximum concentration >7 µg L^−1^). Elevated microcystin concentrations were not positively correlated with numbers of cyanobacterial cells, but multiple linear regression analysis suggested temperature and dissolved oxygen explained a significant amount of the variability in microcystin across both reservoirs. The presence of a managed fishery in one lake was associated with decreased microcystin levels, suggestive of top down control on cyanobacterial populations. This study supports the need to develop inclusive, multifactor holistic water management strategies to control cyanobacterial risks in freshwater bodies.

## 1. Introduction

Cyanobacteria blooms are a global problem in freshwater ecosystems [1,2,3]. A range of factors have been reported to influence the abundance and likelihood of bloom formation in freshwater systems, notably increased temperature, and nutrient enrichment [4,5,6,7]. A proportion estimated as between 40–70%, of cyanobacteria blooms are reported to occur concomitantly with elevated levels of cyanobacterial toxins (microcystins) [8,9,10]. Microcystins are known to be responsible for toxic events globally, most frequently reported are wild animal, livestock, and pet deaths with numerous accounts in the literature, from both more and less economically developed nations [8,11,12,13,14].

In most countries, control plans for public health risks associated with exposure to cyanobacterial toxins are based on assessments of cyanobacterial cell presence and density in the event of bloom formation. In the United Kingdom, assessments and management recommendations are made by national agencies (the Environment Agency (EA) in England and the Scottish Environmental Protection Agency (SEPA) in Scotland). In both administrations, samples are collected reactively from the water column in response to visual bloom occurrence and cyanobacterial are identified to genus level and cells are counted microscopically to determine the cell density in terms of number of cells per millilitre of water. Samples containing >20,000 cells mL^−1^ trigger actions such as preventative closures or restrictions on usage. The presence of cyanobacterial scums on the water surface automatically indicates the need for responsive action, as scum formation is known to increase the likelihood of adverse health effects by factors of up to 1000 [15] and, in the UK, would typically result in measures to prevent exposure of humans and animals [16]. Systematic, or risk-based routine monitoring of water bodies for cyanobacteria is not undertaken in the UK; consequently the incidence, intensity and seasonality of cyanobacterial blooms is not well known [10,17]. Furthermore, whilst the presence of elevated cyanobacterial cells enables identification of potential risks, toxin production during blooms formation is not certain [8,15]. Turner et al. [10] found that only 18% of samples containing cyanobacterial cells exceeding action state thresholds contained microcystins above the WHO medium health criterion of 20 µg L^−1^ in freshwater bodies in England. Therefore, management actions driven by elevated cyanobacterial cell counts may be unnecessary when blooms are formed from non-toxic species and may have unnecessary detrimental economic impacts. Figure 1 shows the occurrence and magnitude data combined with that of Turner et al. [10]; all data were collected in 2016.

Few studies have examined the prevalence and levels of microcystin toxins and variants globally. In the European Multi-Lake Survey, toxin profile data from 26 European countries from lakes with a history of eutrophication were analysed, together with environmental parameters. The authors reported direct and indirect effect of temperature on toxin concentrations and profiles, concluding that whilst few geographical patterns could be discerned, increasing lake temperatures could drive changes in the distribution of cyanobacterial toxins, possibly selecting for a few toxic species [18]. In a study on the array of microcystins during cyanobacterial blooms in Lake Victoria, Tanzania, East Africa, Miles et al. [19] reported a distinctive, complex toxin profile signature during bloom events which has also been confirmed in Ugandan and Kenyan regions of Lake Victoria [19,20]. In a systematic study to assess microcystins in freshwater lakes in England, Turner et al. [10] revealed complex toxin profiles with occurrence of toxin clusters unrelated to cyanobacterial species and no correlation with environmental parameters. These data are suggestive of complex ecosystems, with levels and signatures of microcystin and variants potentially influenced by geographical range but with the impact of environmental factors unclear.

It has been reported that light intensity, temperature, nutrients, and hydrodynamics influence the occurrence and density of cyanobacterial blooms [8,21]. Several studies have attempted to model cyanobacterial concentrations using meteorological, hydrological, and environmental parameters [21,22,23,24,25]. In most studies, the predictive ability of models with respect to risk management has been limited not least because the relationship between the presence and increase in cyanobacterial cells is not always correlated with an increase in the occurrence of toxins [10,26,27]. Notwithstanding this, Carvalho et al. [24] demonstrated that statistical models applied to phytoplankton data from 134 lakes in the UK could be used to describe lakes that may be susceptible to cyanobacterial blooms events. It is evident that understanding the key environmental drivers that favour cyanobacterial abundance and potentially toxic events would facilitate proactive rather than reactive monitoring and management strategies to reduce the public and animal health risks.

In this study, two freshwater reservoirs were routinely monitored by light microscopy, flow cytometry, and liquid chromatography coupled to tandem mass spectrometry over a 12-month period. Measurements of cyanobacterial cells and a range of biological and chemical factors were examined to explore the potential of providing a predictive tool for water management.

## 2. Results

Water measurements and samples were collected and analysed from the May 16, 2016 until May 31, 2017. As stratification was not observed, the data from each depth were combined to produce an average for each measurement at the time of sampling, except for turbidity (NTU), where the lake bottom measurement was disregarded due to sediment disturbance from the horizontal sampler.

### 2.1. Study Site

Longham Lakes consists of two freshwater reservoirs, used as a nature reserve and recreational fishery within the borough boundaries of Bournemouth (Figure 1 and Figure 2). The two lakes located at national grid reference SZ 06237 98079 are man-made, fed by the River Stour and provide an auxiliary water supply to the Bournemouth-Poole conurbation. The northern lake was completed in 2003, has a perimeter of 1400 m, and an area of 97,000 m^2^. The southern lake is connected to the northern lake; it was completed in 2010, has a perimeter of 2050 m, and an area of 250,000 m^2^. The maximum depth for both lakes is approximately 14 m and they both have an average depth of 2.9 m. Longham Lakes is managed by Bournemouth Water, which is part of South West Water. Lake water chemistry and phytoplankton are constantly monitored, and weekly water samples are taken.

### 2.2. Chemical and Biological Parameters

Table 1 shows data collected over the 12-month study period, mean, medium, maxima and minima for total microcystins, *Microcystis* cells, phycocyanin fluorescence, temperature, turbidity, dissolved oxygen, pH, chlorophyll *a*, *b* and total carotenoids are given for the two lakes. A null hypothesis that no differences between biological and chemical measurements were observable between the two lakes was tested at the *p* = 0.05 significance level using a series of Student’s t-tests. No significant differences between temperature, pH, or turbidity were observed between the two lakes over the study period (*p* > 0.05); however, significant and highly significant differences between the two lakes across the sampling period were observed for dissolved oxygen (*p* < 0.001), chlorophyll *a* and *b* levels (*p* < 0.01, *p* < 0.001), and carotenoids (*p* < 0.001) with dissolved oxygen demonstrating the most difference between lakes.

### 2.3. Identification and Enumeration of Phytoplankton by Light Microscopy

A wide range of phytoplankton genera were identified in both lakes between August 2016 and May 2017, a number of chlorophytes and diatoms were only identified to the class level. *Microcystis* cells were recorded in both lakes, maximum >7000 (lake 1) & >8000 cells mL^−1^ (lake 2); other cyanobacteria included *Anabaena,* maximum >34,000 (lake 1) & >21,000 cells mL^−1^ (lake 2), *Aphanizomenon,* maximum >2500 (lake 1) & >20,000 cells mL^−1^ (lake 2), and *Oscillatoria*, maximum >6000 cells mL^−1^ (lake 1 & 2). The non-cyanobacteria identified were *Asterionella, Euglena, Pediastrum, Scenedesmus, Tabellaria,* and *Volvox* (Figure 3). No correlation was found between cyanobacteria identified and counted by either light microscopy or flow cytometry with microcystins detected (data not shown).

### 2.4. Comparison of Counts of Microcystis Cells by Flow Cytometry and Microscopic Method

In both lakes, counts of *Microcystis* cells by flow cytometry were consistently higher and no zero counts were registered as compared to counts by light microscope (Figure 4). A strong correlation between the two methods was observed in lake 1 when tested with a Pearson product moment correlation (*PC* = 0.763, *p* = 0.001), but correlation was not observed between the two methods in lake 2 (*PC* = −0.048, *p* = 0.864).

### 2.5. Determination of Microcystis Cells and Microcystin Concentrations

Figure 5 shows the *Microcystis* cells and total microcystins measured over the study period at both lakes. *Microcystis* cells were detected in both lakes by flow cytometry throughout the sampling period, increasing in July/August in lake 1 and in August in lake 2. An order of magnitude more *Microcystis* cells were detected in lake 1 than lake 2. Mean *Microcystis* cells in lake 1 were 6874 mL^−1^ with a median of 2826 and range of 251 (23 May2016) to 51,384 mL^−1^ (14 July2016). In Lake 2, mean *Microcystis* cells were 1403 mL^−1^ with a median of 1012 and range of 258 (19 December 2016) to 12,204 mL^−1^ (07 March 2017) (Table 1).

Microcystin variants were detected in both lakes but were consistently lower in lake 1 than lake 2. Total microcystin variants and quantities detected are shown in Figure 5. In total, 7 microcystin variants were detected in Lake 1. These comprise of MC-LR, MC-LA, MC-LY, MC-LF, MC-LW, MC-YR, and Asp^3^ MC-LR/[Dha^7^] MC-LR. The microcystin variant detected at the highest concentrations at lake 1 was MC-LF in June and July. Six microcystin variants were detected in Lake 2 (MC-LR, MC-RR, MC-LA, MC-LY, MC-YR and Asp^3^ MC-LR/[Dha^7^] MC-LR) (Figure 6). In Lake 2, MC-YR was detected at highest concentrations during August and September; similar, but slightly lower levels of variant MC-LR was detectable between August and October (Figure 6). Mean total microcystins were 0.5 µg L^−1^ (<LOD to 1.9 µg L^−1^) in lake 1 and 1.5 µg L^−1^ (<LOD to 7.1 µg L^−1^) in lake 2. Maximum levels in Lake 1 were detected between 23 May 2016 and 14 July 2016; microcystins were rarely detected between 3 August 2016 and 28 November 2016. Maximum levels in Lake 2 were detected between 3 August 2016 and 28 September 2016; microcystins were rarely detected between 23 May 2016 & 17 July 2016 and between 3 November 2016 and 31 May 2017.

A null hypothesis that no differences between *Microcystis* cells, total microcystins, and phycocyanin (cyanobacteria cells) measurements were observable between the two lakes was tested using a series of Student’s *t*-tests. Total microcystins and *Microcystis* cells were significantly and highly significantly different between the two lakes respectively (*p* < 0.01, *p* < 0.001) (Table 1).

### 2.6. The Ability of Chemical and Biological Parameters to Predict Presence of Microcystins

Regression analysis suggested that lake, temperature, and dissolved oxygen explained a significant amount of the variability observed in the microcystin values across the two lakes and study period (adjusted R-squared: 0.251, F-statistic: 5.68 on 3 and 39 DF, *p*-value 0.003). Table 1, model 1 predicted microcystin values in to be 2.18 times higher in lake 2 than in lake 1, and that microcystin levels increase with temperature and decrease with increases in dissolved oxygen. No statistically significant interaction terms were observed between any of these variables. Examination of diagnostic plots showed the model fit to be poor, with a high number of zero values observed for microcystin being highly influential on the model fit. No standard data transformations (e.g., log, square root, quadratic) improved the model fit, so a further model (Table 2, model 2) was fit to the subset of the data for which microcystin was detected (i.e., microcystin > 0). The stepwise model building process resulted in the same variables being selected for inclusion in model 2 as for model 1; however, a larger amount of the variability in the observed microcystin values (adjusted R-squared: 0.566, F-statistic: 9.275 on 3 and 16 DF, *p*-value < 0.001) was now explained and examination of the diagnostic plots associated with model 2 showed improved fit. As expected, given the influence of the zero microcystin values in model 1, the effect size associated with the explanatory variables is considerably different in model 2, increasing by around 1.5 times in all cases. Subsequent logistic regression analysis was not able to identify any variables that showed a statistically significant association with the presence/absence of microcystin (i.e., the occurrence of microcystin above the limits of detection).

## 3. Discussion

In this study, relationships between biological and chemical parameters, cyanobacterial taxa with specific reference to *Microcystis* spp., using microscopy and flow cytometry were examined in two lowland lakes in southern Britain. Total microcystins, microcystin variants and toxin profiles were also determined using ultra-high-pressure liquid chromatography coupled to mass spectrometry. *Microcystis* spp. and microcystins were detected and quantified in both lakes. The objective of the study was to attempt to identify drivers of microcystin elevation with a view to better informing water risk management strategies to protect ecological and human health.

Levels of microcystin variants were not correlated with numbers of *Microcystis* cells and were significantly higher in lake 2 than in lake 1. Microcystin levels did not exceed the WHO medium health threshold of 20 µg L^−1^. In 2017, the WHO [15] published chronic health threshold levels of 1 µg L^−1^ MC-LR for lifelong drinking water consumption; 16% of samples from lake 2 exceeded the MC-LR chronic threshold value. Quantifiable levels of toxins were detected in 48% of the samples; identical toxin detection frequencies were observed, but total levels and profiles differed significantly between the lakes. Toxin concentrations ranged from not detected to 7.1 µg L^−1^ and concentrations were similar to those reported by several authors for waterbodies in the absence of scums in the Lower Great Lakes [28], England, and Wales [10] and selected European water bodies [18].

In lake 1, total microcystins were approximately an order of magnitude lower than in lake 2, despite proportionally higher isolation frequencies and levels of *Microcystis* cells (lake 1, maximum density 51,384 cells mL^−1^). *Microcystis* cells did not exceed 8000 cells mL^−1^ in lake 2 and constituted a relatively minor fraction of the total estimates of cyanobacteria. It is probable that *Microcystis* cells at both lakes were non-toxin producing strains and that toxins detected in this study were produced by species other than *Microcystis.* It is well documented that the *Microcystis* blooms vary in their toxin profiles [10,15,29]. The ability for microcystin production in *Microcystis* spp. and other cyanobacterial species is genetically determined [30,31]. Several studies have reported that strains isolated from geographically and temporally distinct *Microcystis* spp. populations are clonal and therefore likely to be either microcystin producers or non-producers [5,32]. The hypothesis that species other than *Microcystis* spp. were responsible for toxin production in the study lakes is supported by the presence of other cyanobacteria, e.g., *Anabaena* spp., *Aphanizomenon* spp., and *Oscillatoria* spp. at elevated levels particularly in lake 2. At lake 2, on the 3 occasions where the UK threshold action limits were exceeded, less than 8.5% of the estimated cyanobacterial populations comprised *Microcystis* cells. In a review of cyanobacterial bloom taxa in the UK, Howard et al. [33] recorded that the dominant species in addition to *Microcystis* spp., were *Oscillatoria, Planktothrix, Anabaena, Pseudanabaena,* and *Gomphosphaeria.* The dominance of these species amongst phytoplankton communities in samples from natural lakes and reservoirs has been subsequently confirmed in England and Wales [10,34] and Scotland [17]. *Anabaena* spp., *Aphanizomenon* spp., and *Oscillatoria* spp., are well known as toxin-producing species and common members of phytoplanktonic communities [29].

Maximum total microcystin level recorded was in lake 2 (7.1 µg L^−1^), with mean levels in lake 1 of 0.497 µg L^−1^ and 1.524 µg L^−1^ in lake 2. Total microcystins and *Microcystis* cells were significantly and highly significantly different between the two lakes, respectively (total microcystins higher in lake 2; *Microcystis* spp. cells higher in lake 1). Of the multiple microcystin variants described, MC-LR has been most extensively studied and is reported to be between 3 and 10 times more toxic than other microcystin congeners [28]. Analysis of the microcystin toxin profiles between the two study lakes indicated differences in variants, both in terms of variants, proportions, and levels. In lake 1, where relatively low levels of microcystins were determined, MC-LR was detected at low levels (<1 µg L^−1^) and the dominant variant was the more hydrophobic MC-LF. This finding is in accordance with Turner et al. [10], who reported MC-LF as the highest mean proportion of profiles from *Aphanizomenon* sp. and *Oscillatoria* sp. in analyses of freshwater bodies in England and Wales. This adds some support to the premise that *Microcystis* cells present in lake 1 were non-toxin producers. In lake 2, MC-YR was the dominant congener closely followed by MC-LR, with MC-RR and Asp^3^-MC-LR appearing towards the end of the period of toxin prevalence. Microcystin profiles determined in samples taken from weekly monitoring during August and September were similar in relative proportions, indicating potentially that clonal or semi-clonal populations were responsible for toxin production. Similar data have been generated from studies in Greece [35], throughout Europe via the European Multi Lake Survey [18] and Finland [36], the latter associated with toxins produced by species of *Anabaena.* Whilst data on the relative proportions of microcystin variants as measured by LC-MS/MS are relatively sparse, levels and variants of microcystins presented here are consistent with the recorded literature and can help to unravel the structure and function of cyanobacterial populations.

In recent years, many authors have studied the drivers for elevated cyanotoxin levels and/or occurrence of cyanobacterial blooms in natural lakes, reservoirs [18,37,38,39], and aquaculture systems [40]. In a large-scale study assessing the continental scale distribution of cyanotoxins across Europe, Mantzouki et al. [18] demonstrated that temperature (rather than nutrient availability or euphotic depth) was generally responsible for distributional characteristics of cyanotoxins. Likewise, concordant observations with respect to temperature were recorded by Elliot [39] in an assessment of climate change on pelagic freshwater cyanobacteria. The author demonstrated increased relative cyanobacteria abundance concurrently with increased water temperature, together with decreased flushing rates and increased nutrient loading. Similarly, Sinden and Sinang [40] identified temperature, in combination with elevated water pH, as key environmental factors influencing proliferation of cyanobacteria and toxicity in Malaysian aquaculture ponds. In the present study, the presence of correlations between a range of biological and chemical parameters were tested against presence of microcystins. Perhaps not surprisingly given the low levels of microcystins detected in lake 1, no correlations were observed. For lake 2, where moderate levels of microcystins were present during the summer and autumn, water temperatures did not correlate; however, a decrease in dissolved oxygen was closely associated with presence and levels of microcystins. A decrease in dissolved oxygen concurrent with decomposition of cyanobacterial blooms has been reported previously [6,41] and thus has the potential to indicate onset of toxicity derived from lysed cells within a rapidly blooming population. Perturbations in dissolved oxygen are frequently used as an indicator of water quality and eutrophication [42]. Conversely, with respect to the correlations of microcystin production, in a comprehensive review of biological and chemical factors, Dai et al. [42] postulated that light intensity and temperature were the most important physical factors with nitrogen and phosphorus as the critical chemical drivers of harmful algal blooms and microcystins. The authors also noted the complex interactions with biotic factors, suggesting that predator-prey relationships in phytoplanktonic communities may promote microcystin production and release [42].

To explain the strength of the association between the measured parameters and their potential future ability to inform predictive models for microcystin events, stepwise multiple linear regression analysis was applied to all chemical and biological parameters. Across both lakes, microcystins increased with temperature and decreased with dissolved oxygen. Using a similar statistical approach as a precursor to the development of empirical predictive models for cyanobacterial, Beaulieu et al. [43] showed that total nitrogen and water temperature provided the best model and explained 25% of cyanobacterial biomass. Using these explanatory variables, the authors developed competing path models, which showed that both nitrogen and temperature were indirectly (and directly) linked to cyanobacteria by interactions with total algal biomass. Model outputs predicted an average doubling of cyanobacterial biomass with a 3.3 °C rise in water temperature. In contrast, Carvalho et al. [24] showed that significant explanatory variables were dissolved organic carbon and pH, and that furthermore nutrient concentrations were not a primary explanatory variable. In this study the relatively high levels of variability explained by combined biological and chemical parameters via regression analysis indicated promise and areas for future investigation.

Although in this study, no single variable could be considered predictive of microcystin production nor could all the variability between the lakes be explained by combinations of measurements, the major identifiable difference between the two water bodies was the presence of a managed fishery at lake 1. Omnivorous fishes; common carp (*Cyprinus carpio*), bream (*Abramis brama*), tench (*Tinca tinca*) and pike (*Esox lucius*), and substantial natural populations of roach (*Rutilus rutilus*), and rudd (*Scardinius erythrophthalmus*) were present in lake 1, whereas no managed fishery or introductions of fish species operated at lake 2. Larval, fry, and fingerlings of stocked species favour zooplankton, with certain high nutrient species (rotifers etc.) making up a substantial component of adult fish’s food supply. Larval stages and young age classes of *R. rutilus* and *S. erythrophthalmus,* numerous during summer months are almost exclusively zooplanktoniverous, with a feeding preference for rotifers [44]. Rotifers and other small zooplankton, such as cyclopoid copepods and cladocerans, are selective grazers that can coexist with bloom forming cyanobacteria and are reported to periodically exhibit top-down control [45]. Interestingly, several authors have suggested that *Microcystis* spp. represent a less attractive foodstuff for zooplankton due inter alia to colony formation [45,46,47], which together with a reduction in grazing pressure on other cyanobacterial taxa may have created a competitive advantage for the non-toxic *Microcystis*-like cell populations, such as unknown species of picoplankton in lake 1. *R. rutilus* abundance has previously been implicated in ichthyo-eutrophication of reservoirs in the Czech Republic [44]; this has been attributed to a range of factors but may also include reduction in grazing rates by zooplankton, which in turn create advantageous conditions for blooms [48]. Furthermore, eutrophication can change microbial loops and therefore may inhibit antagonistic microorganisms (viruses, bacteria, microalgae, microfungal, and amoeboid taxa) present within *Microcystis* colonies in the operation of bottom-up controls [45].

Environmental drivers for toxin production from cyanobacterial taxa are complex and intricacies of phytoplankton communities, cryptic ecological interactions, and the presence of non-toxin producing cyanobacterial strains make this a challenging area of risk management. This study demonstrated that no single variable could be used to predict microcystin levels but supported the use of multiple measurements in the development of more holistic predictive models. In this study, multiple measurements were dissolved oxygen, turbidity, phycocyanin, temperature pH, chlorophyll *a, b,* and total carotenoids. The potential for enhanced fish stocks to exert top down control on toxic cyanobacterial populations was an interesting observation and is indicative of wider multifaceted trophic relationships influencing cyanobacterial population dynamics and toxin production. Whilst challenging at an environmentally relevant scale [21], improved predictive ability and modelling will provide more efficient, proactive management of water bodies impacted by toxic cyanobacteria and in turn have positive ecological, public, and animal health benefits.

## 4. Materials and Methods

### 4.1. Sample Collection and Water Parameter Measurements

Water samples were collected with chemical and biological measurements between May 16 2016 and May 31 2017, from both northern and southern lakes at points A & B (Figure 2). The lakes were sampled weekly in the spring, summer, and autumn, with sampling frequency dropping to every three weeks in the winter. Measurements of dissolved oxygen, turbidity, phycocyanin, temperature, pH, and salinity, were made by multiparameter probe (6600 V2, YSI, Xylem Analytics, Singapore) from the surface and at 1 m intervals to the bottom. Water samples for flow cytometer and toxin analysis were collected from the same depths using a 2.2 L horizontal sampler. Water samples for microscope analysis were collected from the surface of each lake only. Water samples were in put in opaque bottles and stored in a cool box for return to the laboratory.

### 4.2. Cell Discrimination by Flow Cytometry

Water samples were analysed for using a flow cytometer (C6, BD Accuri, San Jose, CA, USA), samples were aliquoted (≈2 mL) into a 5 mL sample tube, and homogenised by vortex. A 5-min custom fluidic setting was selected of 25 μm core and 100 μL/min, with a threshold of 80,000 au on forward scatter (FSC) signal. Unicellular *Microcystis* cells were resolved by size and three auto-florescence channels. Following Chapman [49], side scatter (SSC) signal was used as an indicator cell size and verified by calibration beads (PPS-6K, Spherotech, Lake Forest, IL, USA), gated between 75,000 to 700,000 au. Yellow auto-florescence (FL2) was used as an indicator of phycoerythrin and carotenoids, gated at 40 to 2000 au. Red auto-florescence (FL3) was used as an indicator of chlorophyll, gated between 180,000 to 2,200,000 au. Finally, far-red auto-florescence was used as an indicator of phycocyanin, gated between 11,000 to 500,000 au. The method was optomised using 6 *Microcystis*, 2 non-*Microcystis* cyanobacteria, and 2 eukaryotic algal reference strains.

### 4.3. Toxin Analysis by Liquid Chromatography Coupled to Tandem Mass Spectrometry

In triplicate, 200 mL of water sample was filtered (CFC, Whatman, Maidstone, UK), filter papers were wrapped individually in aluminium foil and preserved at −80 °C. On analysis, filter papers were subjected to three cycles of freeze-thawing before submersion in 10 mL of 80% aqueous methanol. Samples were left in the dark at 4–6 °C for 24 h, before ~0.5 mL was aliquoted into a LCMS certified vial. Toxin analysis was carried by ultra-high-performance liquid chromatography (UHPLC) (Acquity, Waters, Manchester, UK) coupled to a tandem quadruple mass spectrometer (Xevo TQ, Waters, Manchester, UK). All instrument solvents and chemicals were of LC-MS-grade (Fisher Optima, Thermo Fisher, Manchester, UK). Reference toxins used for the detection method included the microcystin analogues MC-RR, MC-LA, MC-LY, MC-LF, MC-LW, MC-YR, MC-WR, MC-HilR, MC-HtyR, MC-LR & Asp^3^-MC-LR (Enzo Life Sciences, Exeter, UK) and [Dha^7^]-MC-LR and matrix reference material of blue-green algae (RM-BGA, Lot 201301) containing a range of microcystins (Institute of Biotoxin Metrology, National Research Council Canada). Analysis of microcystins was conducted following the method by Turner et al. [50].

### 4.4. Identification and Enumeration of Phytoplankton by Light Microscopy

Surface water samples were aliquoted into a 15 mL centrifuge tube; the full tube was sealed then inverted and the lid struck on the bench several times to burst any gas flotation vesicles within cells. Centrifuge tubes were then placed upright and stored at 4–6 °C in the dark for two days for the phytoplankton to settle out. The top 14 mL was carefully removed by pipette to not disturb the sedimented phytoplankton; the remaining 1 mL was vortexed and transferred to a Sedgwick-rafter counting chamber. The phytoplankton in a minimum of 10 of the 1000 grid squares were identified and enumerated by light microscope (BX51, Olympus, Tokyo, Japan) at 40× and 100× magnification.

### 4.5. Multiple Linear Regression Model

Relationships between the presence and level of microcystin and potential predictor variables were explored visually using plot functions and the strength of these relationships assessed using multiple linear regression models after making appropriate transformations data (if required) to ensure its distribution met the test assumptions. A logistic regression model (GLM assuming a binomial distribution and log link function) was also applied to determine whether any of the environmental variables measured were able to reliably predict the presence/absence of microcystin rather than its levels. For both the linear and logistic regression analysis, univariable analysis was first performed and then a multivariable model was built using a forward stepwise approach in which variables and combinations that led to statistically significantly reductions (*p* ≤ 0.05) in the Akaike information criteria (AIC) were retained in the model. The presence of two-way interaction effects was explored for all variable combinations but were only retained if their inclusion resulted in a significant reduction in AIC. All analysis and data visualisations were conducted in R version 4.0.2 [51].

## Figures and Tables

**Figure 1 toxins-12-00503-f001:**
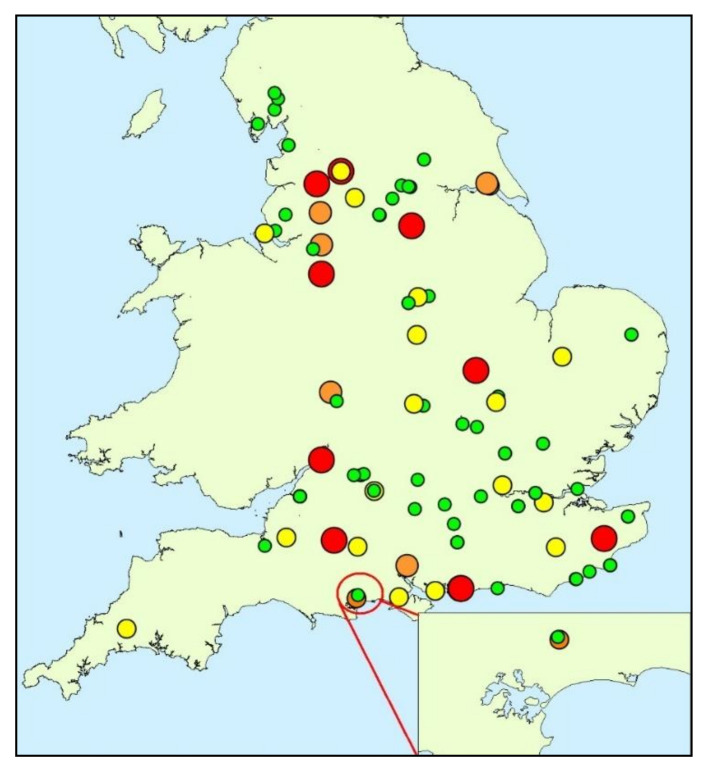
Occurrence and magnitude of total microcystins recorded from England and Wales in 2016. (red: >100 µg/L; orange: 20–100 µg/L; yellow: 2–20 µg/L; green: <2 µg/L). Insert, microcystin data and location of the study site, Longham Lakes, Bournemouth, Dorset, UK. (adapted from Turner et al. [10]).

**Figure 2 toxins-12-00503-f002:**
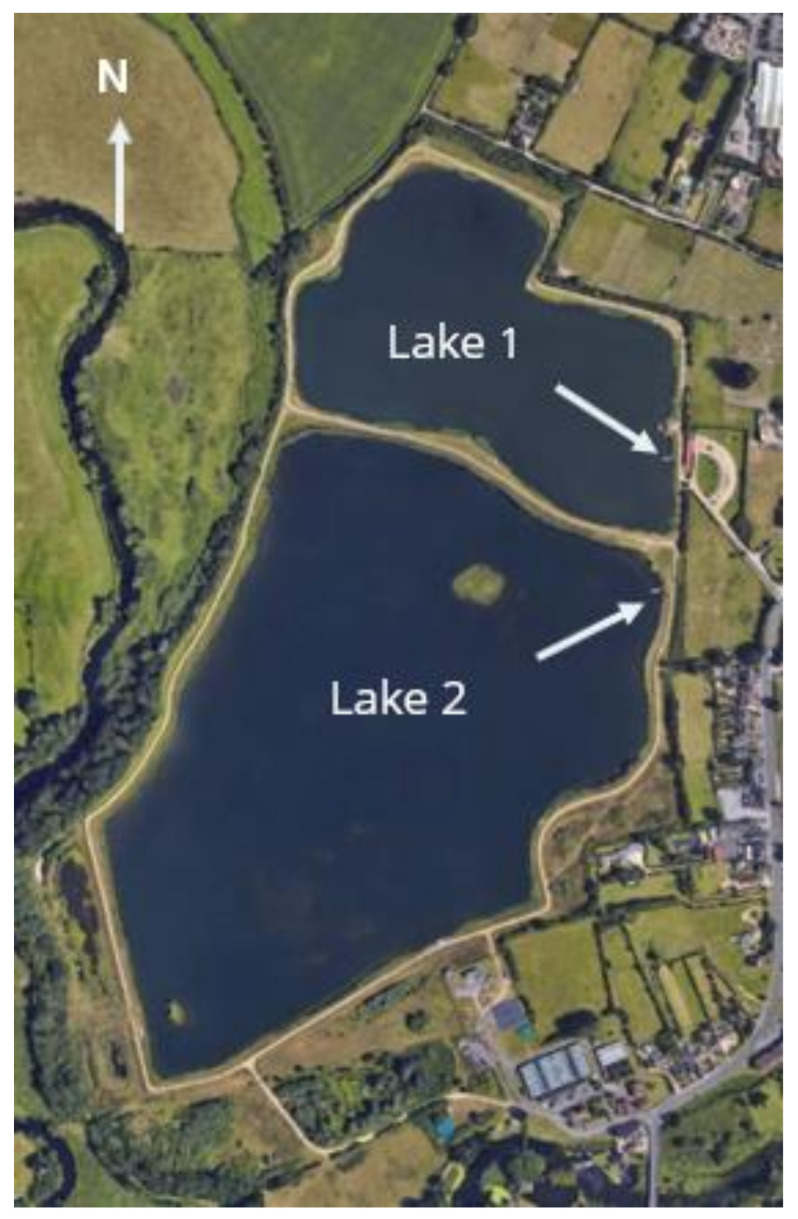
Aerial view of Longham Lakes with sampling point marked by arrows in Lake 1 (northern) and Lake 2 (southern).

**Figure 3 toxins-12-00503-f003:**
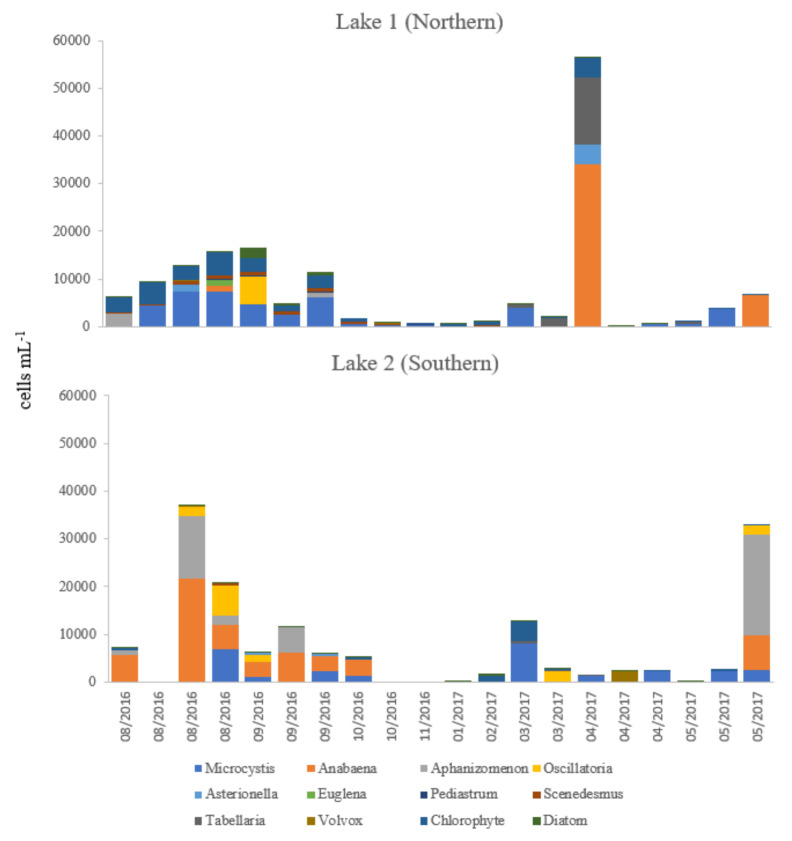
Stacked bar chart showing the date, number, and taxa of phytoplankton identified in Longham Lakes 1 & 2 by light microscope.

**Figure 4 toxins-12-00503-f004:**
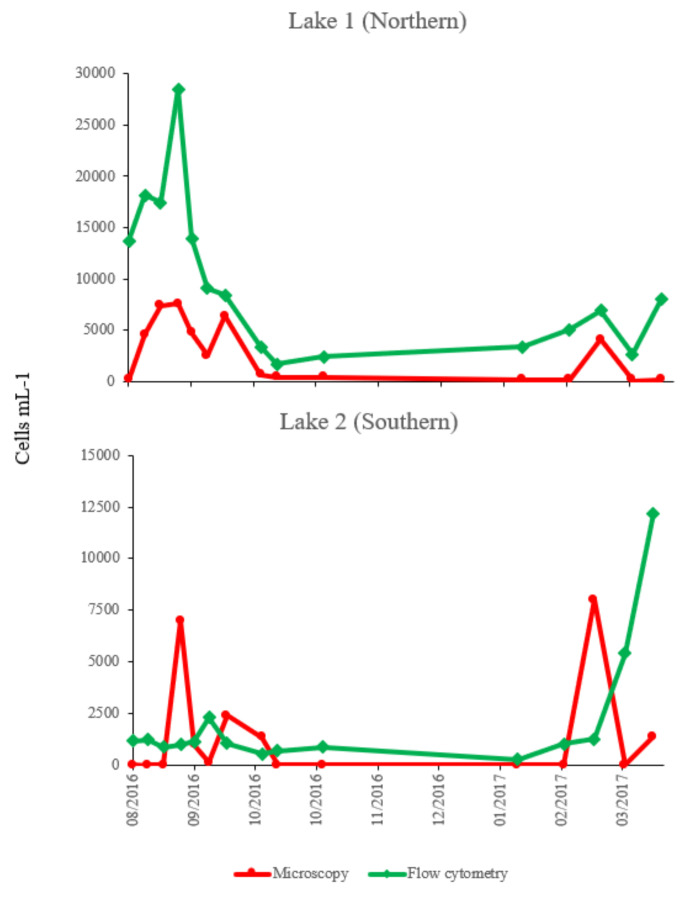
Comparison of counts of *Microcystis* cells in both lakes at Longham, as counted by flow cytometry and microscope methods over the study period.

**Figure 5 toxins-12-00503-f005:**
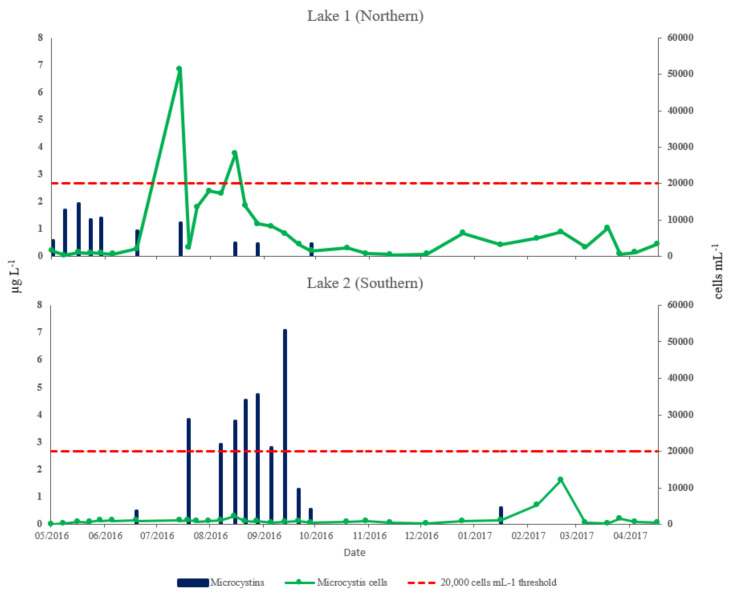
Seasonal variation recorded at Longham Lake (1 & 2) of *Microcystis* cells (cells mL^−1^) by flow cytometry (right-hand axis) and total microcystins quantified by liquid chromatography coupled to tandem mass spectrometry (µg L^−1^) (left-hand axis). Red line indicates UK cyanobacterial cell density action threshold [16].

**Figure 6 toxins-12-00503-f006:**
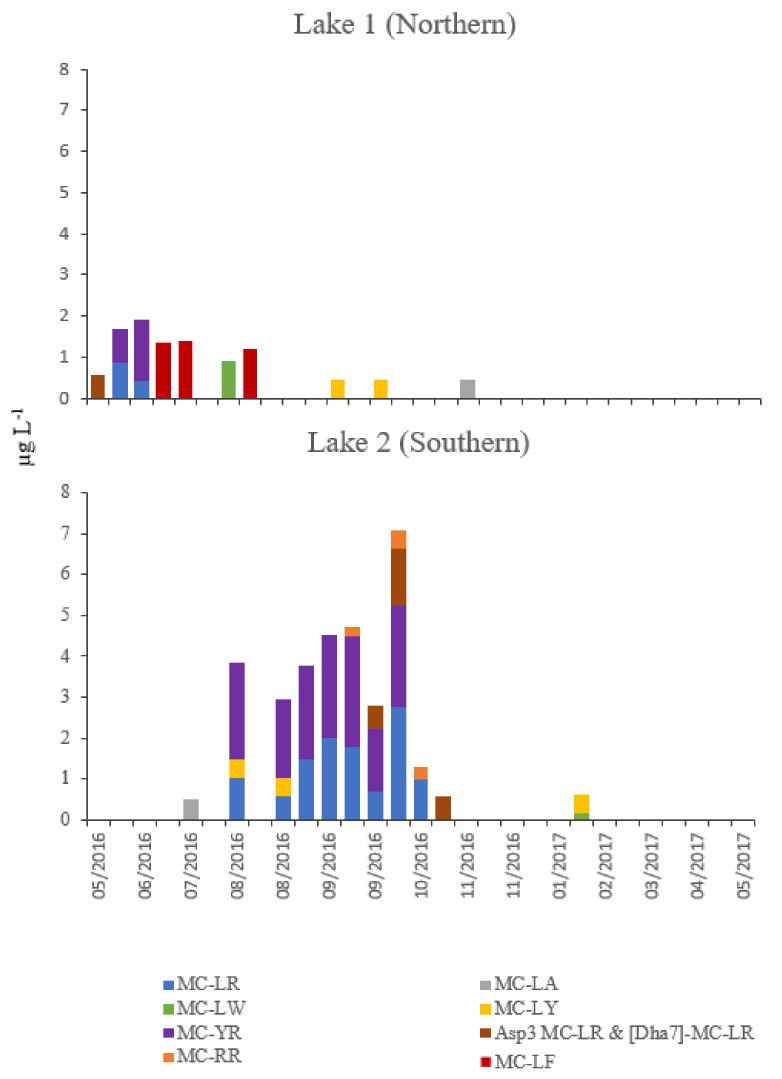
Microcystin variants qualified and quantified by liquid chromatography coupled to tandem mass spectrometry (µg L^−1^) from water samples collected at Longham Lakes (1 & 2).

**Table 1 toxins-12-00503-t001:** Biological and chemical measurements from Longham Lakes 1 and 2, between 16 May 2016 and 31 May 2017.

Parameter	Lake 1 (Northern)				Lake 2 (Southern)				Student
	Low	Mean	Median	High	Low	Mean	Median	High	*t*-Test
Total microcystins (µg L^−1^)	nd ^1^	0.497	nd ^1^	1.922	nd ^1^	1.524	nd ^1^	7.089	*p* < 0.01
*Microcystis* cells (cells mL^−1^)	251	6874	2826	51,384	258	1403	1012	12,204	*p* < 0.001
Phycocyanin (Cells mL^−1^)	109	1425	836	7649	20	1924	705	10,290	*p* > 0.05
Temperature (°C)	5.57	14.96	16.51	21.64	5.81	15.13	15.88	21.51	*p* > 0.05
Turbidity (NTU)	−0.40	2.25	1.50	8.90	−1.60	1.51	0.70	8.20	*p* > 0.05
Dissolved Oxygen (mg L^−1^)	6.14	12.19	12.07	24.16	9.44	12.74	12.99	18.98	*p* < 0.001
pH	7.52	8.44	8.47	9.29	8.06	8.52	8.54	8.97	*p* > 0.05
Chlorophyll *a* (mg mL^−1^)	0.44	3.821	2.398	15.373	0.042	1.315	0.969	4.056	*p* < 0.01
Chlorophyll *b* (mg mL^−1^)	0.41	2.296	2.206	6.752	nd	1.294	1.143	4.367	*p* < 0.001
Total Carotenoids (mg mL^−1^)	nd	1.200	0.676	6.295	nd	0.260	0.135	1.467	*p* < 0.001

^1^ Limit of detection (LOD) for MC-LR = 0.0013 ± 0.0011 ng mL^−1^ [10].

**Table 2 toxins-12-00503-t002:** Multiple linear regression predictors for microcystin levels in both lakes. Model 1 includes observations where microcystin was not detected (i.e., microcystin = 0), model 2 shows results for data relating to positive microcystin observations only (i.e., microcystin > 0).

Parameter	Estimate	Std. Error	*t*-Value	Pr(>|*t*|)
Model 1: Zero microcystin values included (*n* = 43)				
Intercept	1.446	1.295	1.116	0.271
Lake	2.183	0.583	3.748	0.001
Dissolved O_2_	−0.418	0.143	2.920	0.006
Temperature	0.173	0.062	2.780	0.008
Model 2: Zero microcystin values removed (*n* = 20)				
Intercept	1.408	2.358	0.597	0.559
Lake	3.523	0.698	5.051	0.000
Dissolved O_2_	−0.596	0.194	3.076	0.007
Temperature	0.295	0.119	2.470	0.025

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
