# Peer review of "Cyanobacterial Abundance and Microcystin Profiles in Two Southern British Lakes: The Importance of Abiotic and Biotic Interactions"

_toxins, 2020, doi:10.3390/toxins12080503_

Round 1

Reviewer 1 Report

The paper “Cyanobacterial Abundance and Microcystin Profiles in Two Southern British Lakes: The Importance of Abiotic and Biotic Interactions” deals with the relationships between biological and chemical parameters, cyanobacterial taxa in two lakes in southern Britain, including total microcystins and microcystin variants analysis, with the aim to attempt to identify drivers of microcystin presence for water risk management strategies and ecological and human health protection. In my opinion it is well written and very interesting. The differences between the two lakes in terms of microcystins and cyanobacterial composition are very interesting, as well the conclusions about the importance of considering multiple measurements in predictive models for microcystin levels. At the same time, considerations about the effect of a managed fishery present in one of the lake on the trophic relationships open new insights in the importance of considering several factors to understand algal population dynamics.

I have some questions and suggestions:

Page 5 line 136-138: this sentence is not clear “No correlation was found…”

Did authors find any correlation between the number of cells and the chlorophyll amount in water samples?

What about isolating single cyanobacterial cells in order to set up cultures? Authors should considered it to attest the toxicity of the different species, in particular the non-toxicity of Mycrocystis and potential toxicity of other species, as supposed in the discussion.

I would suggest some minor corrections:

Figure 3 – I would suggest to add “by light microscopy” in the caption

Figure 5 – please correct “left” in “total microcystins quantified 160 by liquid chromatography coupled to tandem mass spectrometry (μg L−1) (left-hand axis)”

Figure 6 – please add “by” in “Microcystin variants qualified and quantified by liquid chromatography”

Page 11 line 219: delete the round bracket after species

Page 12 line 275: correct “the most important physical factors”

Page 13 line 321: Please delete “the” in “In this study..”

Page 13 line 350: correct “optimised”

M&M

A detailed paragraph describing the statistical methods and analyses is missing. Please add details about the regression model used.

Reviewer 2 Report

Cyanobacterial abundance and Microcystin profiles in two southern British Lakes: The importance of Abiotic and Biotic Interaction

This manuscript profiles cyanobacteria and microcystins measurements by flow cytometry and liquid chromatography in two lakes in southern Britain over a 12-month period. Based on multiple linear regression analysis, the authors concluded that the elevated microcystin concentrations were not correlated with numbers of cyanobacterial cells in either lake, and neither microcystin concentration nor cyanobacterial cell numbers can be predicted by single abiotic or biotic variables.

It is meaningful to profile the cyanobacteria and microcystin measurements in the lakes. However, the analysis based on the multiple linear regression analysis is superficial. The authors stated that the combined independent variable explained approximately 60-65% of the variability at lakes 1 and 2 respectively. 1). What is your multiple linear regression model? Please at least list the parameter vector. 2) I suggest you comparing the time series of predicted value with the ground truth (real observations), which would provide more information about the how the model explained the temporal variability. 3) Some abiotic/biotic variables may have no linear relationship with the cyanobacteria/microcystin concentrations at all. This makes the linear regression analysis less meaningful. In addition, I don’t see significant difference between the R2 values for the multiple linear regression analysis based on all variables and some independent variables, especially in Lake 2. Therefore, my interpretation of the analysis results is that the cyanobacteria/microcystin concentration may not have linear relationship with the abiotic/biotic variables listed. Based on the facts, I don’t think the study can better inform water risk management strategies to protect ecological and human health. Nevertheless, as I emphasized at the beginning, the study is meaningful to monitor and profile the observations in the two lakes.
